# The pluripotent stem cell-specific transcript ESRG is dispensable for human pluripotency

Kazutoshi Takahashi[1,2]*, Michiko Nakamura[1], Chikako Okubo[1], Zane Kliesmete[3], Mari Ohnuki[3], Megumi Narita[1], Akira Watanabe[4], Mai Ueda[1], Yasuhiro Takashima[1], Ines Hellmann[3], Shinya Yamanaka[1,2,5]

1 Center for iPS Cell Research and Application, Kyoto University, Kyoto, Japan, 2 Gladstone Institute of Cardiovascular Disease, San Francisco, California, United States of America, 3 Anthropology and Human Genomics, Department of Biology II, Ludwig-Maximilians Universitaet, Munich, Germany, 4 Graduate School of Medicine, Kyoto University, Kyoto, Japan, 5 Department of Anatomy, University of California, San Francisco, San Francisco, California, United States of America

* kazu@cira.kyoto-u.ac.jp

**Data Availability Statement:** The numerical values for the graphs in the manuscript are provided as the Supporting Data. RNA-seq (GSE56568 and GSE171849), ChIP-seq (GSE56567 and

## Abstract

Human pluripotent stem cells (PSCs) express human endogenous retrovirus type-H (HERV-H), which exists as more than a thousand copies on the human genome and frequently produces chimeric transcripts as long-non-coding RNAs (lncRNAs) fused with downstream neighbor genes. Previous studies showed that HERV-H expression is required for the maintenance of PSC identity, and aberrant HERV-H expression attenuates neural differentiation potentials, however, little is known about the actual of function of HERV-H. In this study, we focused on ESRG, which is known as a PSC-related HERV-H-driven lncRNA. The global transcriptome data of various tissues and cell lines and quantitative expression analysis of PSCs showed that ESRG expression is much higher than other HERV-Hs and tightly silenced after differentiation. However, the loss of function by the complete excision of the entire ESRG gene body using a CRISPR/Cas9 platform revealed that ESRG is dispensable for the maintenance of the primed and naïve pluripotent states. The loss of ESRG hardly affected the global gene expression of PSCs or the differentiation potential toward tri-lineage. Differentiated cells derived from ESRG-deficient PSCs retained the potential to be reprogrammed into induced PSCs (iPSCs) by the forced expression of OCT3/4, SOX2, and KLF4. In conclusion, ESRG is dispensable for the maintenance and recapturing of human pluripotency.

## Author summary

We have been interested in the role of human endogenous retrovirus (HERVs) in human pluripotent stem cells (PSCs). Although we and others have demonstrated that HERV expression is crucial for somatic cell reprogramming to a pluripotent state and the characteristics of PSCs. Little is known which one of more than 1,000 copies of HERVs is important. Thus, in this study, we focused on a HERV-related gene, ESRG which is expressed strongly and specifically in human PSCs but not in differentiated cells. Using a

GSE89976) and Gene expression microarray (GSE54848, GSE156834, GSE159101, and GSE171627) results are accessible in the Gene Expression Omnibus database of the National Center for Biotechnology Information website.

**Funding:** This work was supported by Grants-in-Aid for Scientific Research (20K20585) to K.T. from the Japanese Society for the Promotion of Science (JSPS); a grant from the Core Center for iPS Cell Research (JP21bm0104001) to S.Y., Research Center Network for Realization of Regenerative Medicine from Japan Agency for Medical Research and Development (AMED) to S. Y.; a grant from the Japan Foundation for Applied Enzymology to K.T.; a grant from the Fujiwara Memorial Foundation to K.T.; a grant from the Takeda Science Foundation to K.T.; and the iPS Cell Research Fund to K.T. from Center for iPS Cell Research and Application, Kyoto University. The study was also supported by funding to S.Y. from Mr. H. Mikitani, Mr. M. Benioff, and the L.K. Whittier Foundation. The funders had no role in study design, data collection and analysis, decision to publish, or preparation of the manuscript.

**Competing interests:** I have read the journal's policy and the authors of this manuscript have the following competing interests: K.T. is on the scientific advisory board of I Peace, Inc. without salary, S.Y. is a scientific advisor (without salary) of iPS Academia Japan, and other authors have declared that no competing interests exist.

CRISPR/Cas9 platform, we generated complete knockout cell lines by deleting the entire gene body of ESRG.

Our results demonstrate that ESRG is dispensable for the PSC characters such as gene expression, self-renewing capacity, and differentiation potential. In addition, ESRG does not contribute to the reprogramming of differentiated cells to a pluripotent state. Altogether, we concluded that ESRG is an excellent marker of pluripotency but dispensable for the PSC identity.

## Introduction

Human pluripotent stem cells (PSCs) express several types of human endogenous retroviruses (HERV) [1–3]. The HERV type-H (HERV-H) family is a primate-specific ERV element that was first integrated prior to the New World/Old World divergence. During further primate evolution, this family's major expansion occurred after the branch of Old World monkeys [4]. The typical structure of a HERV-H consists of an interior component, HERV-H-int, flanked by two long terminal repeat 7 (LTR7), which have promoter activity [5,6]. Recent studies have demonstrated that the activity of LTR7 is highly specific in established human PSCs and relatively absent in early human embryos. In contrast, other LTR7 variants such as LTR7B, C, and Y are activated in broad types of early human embryos from the 8-cell to epiblast stages [7].

The importance of HERV-Hs in human PSCs has been shown. The knockdown (KD) of pan HERV-Hs using short hairpin RNAs (shRNAs) against conserved sequences in LTR7 or HERV-H-int regions revealed that HERV-H expression is required for the self-renewal of human PSCs [8,9] and somatic cell reprogramming toward pluripotency [8–14]. In addition to self-renewal, the precise expression of HERV-Hs is crucial for the neural differentiation potential of human PSCs [10,15]. In this way, HERV-H expression contributes to the PSC identity.

The transcription of HERV-H frequently produces a chimeric transcript fused with a downstream neighbor gene, which diversifies HERV-H-driven transcripts. Therefore, many HERV-H-driven RNAs contain unique sequences aside from HERV-H consensus sequences. Indeed, PSC-associated HERV-H-containing long non-coding RNAs (lncRNAs) have been reported [15–17]. One of them, ESRG (embryonic stem cell-related gene; also known as HESRG) was identified as a transcript that is predominantly expressed in undifferentiated human embryonic stem cells (ESCs) [18,19]. ESRG is transcribed from a HERV-H LTR7 promoter [8,20] and is activated in an early stage of somatic cell reprogramming induced by the forced expression of OCT3/4, SOX2, and KLF4 (OSK) [12,13,20]. One previous study showed that the shRNA-mediated KD of ESRG induces the loss of PSC characters such as colony morphology and PSC markers along with the activation of differentiation markers, suggesting the indispensability of ESRG for human pluripotency [8]. However, despite these characterizations, the function of ESRG is still unknown.

In this study, we analyzed the conservation of ESRG to infer its functional importance. Then we completely deleted ESRG alleles to analyze ESRG function in human PSCs with no off-target risk. The loss of ESRG, which is thought to be an essential lncRNA for the PSC identity [8], exhibited no impact on the self-renewal or differentiation potentials of both primed and naïve human PSCs. Neural progenitor cells (NPCs) derived from ESRG-deficient PSCs could be reprogrammed into induced PSC (iPSC) by OSK expression. Altogether, this study revealed that ESRG is dispensable for human pluripotency.

## Results

### No evidence for ESRG conservation

A large proportion of the ESRG lncRNA-gene is derived from a HERV-H insertion event that happened after the orangutan split from the other great ape lineages leading to humans and chimpanzees [21]. The entire first exon and part of the second exon of ESRG are encoded by this HERV-H element (Fig 1A). Accordingly, the conservation as determined by PhastCons scores [22,23] is low throughout the transcript (0.7% of sites with PhastCons>0.9), even when compared to other lncRNA-genes (Fig 1A and S1 Table). In humans, chimpanzees, and bonobos, the entire element is present, while in gorilla only partial sequences of the LTR7 flanks are left. However, even though ESRG is present in chimpanzees, it shows a much lower expression in iPSCs than in humans (Fig 1B and S2 Table). As expected, ESRG is highly expressed in iPSCs and then downregulated upon differentiation as can be seen in the iPSC-derived cardio-myocytes [24]. Indeed, in human iPSCs, ESRG is alongside OCT3/4 and GAPDH among the 5% most highly expressed genes but ranks lower than 50% in chimpanzees (S3 Table). Hence, even though ESRG is present in chimpanzees, its expression pattern is not conserved.

However, also transcripts that are not phylogenetically conserved can be of functional importance. Such transcripts should carry signatures of negative selection. If ESRG had an important function in human populations, then we should find signs for deleterious and slightly deleterious alleles which can segregate at low frequencies within a population but are less likely to get fixed [25,26]. Unfortunately, the power to detect negative selection in population genetics data is relatively low, in particular, if only a small proportion of sites is expected to be under selection. For example, only 8% of sites in HOTAIR, a well-documented lncRNA [27] are notably conserved (PhastCons>0.9). To detect deleterious sites, we compared human-chimpanzee divergence of exon and intron sequences and find that divergence in exons is not significantly lower than in the introns of ESRG (Fisher's-Exact test, $d_{exon}/d_{intron}$ = 0.85, p = 0.51; Fig 1C and S4 Table). To detect slightly deleterious sites, we checked for a left shift of the site frequency spectrum [25] and found that the proportion of singletons in ESRG exons is much lower than for the on average highly conserved non-synonymous SNVs and similar to SNVs in other non-coding exons and synonymous sites (Fig 1D). Also compared to other lncRNAs, both conserved and nonconserved, ESRG has no shift towards rare alleles (Fig 1E). Next, we looked for a lower fixation rate of mutations occurring in ESRG exons as compared to introns by contrasting the number of human SNVs [28] with the number of single nucleotide substitutions (SNS) between humans and the common ancestor of chimpanzees and bonobos (Fig 1C). Even though the intronic sequences have a slightly higher fixation rate than the exon the difference is not significant (Fisher's-Exact test, $(SNS_{exon}/SNV_{exon})/(SNS_{intron}/SNV_{intron})$ = 0.74, p = 0.21). All in all, we do not find any compelling evidence for selection.

### ESRG is robustly expressed in human PSCs and tightly silenced after differentiation

To acquire an in-depth understanding as to the ESRG expression in humans, we analyzed the expression and epigenetic statuses of the ESRG gene in human PSCs and human dermal fibroblasts (HDFs). The RNA sequencing (RNA-seq) and chromatin immunoprecipitation sequencing (ChIP-seq) of histone H3 modifications [10] indicated that the ESRG locus is open and actively transcribed in human PSCs but not in differentiated cells such as human dermal fibroblasts (HDFs) (Fig 2A). As well as other HERV-H-related genes, LTR7 elements in the ESRG gene are occupied by pluripotency-associated transcription factors (TFs) such as OSK [9,10] (Fig 2A). Little or no ESRG expression was detected in 24 human adult tissues and five

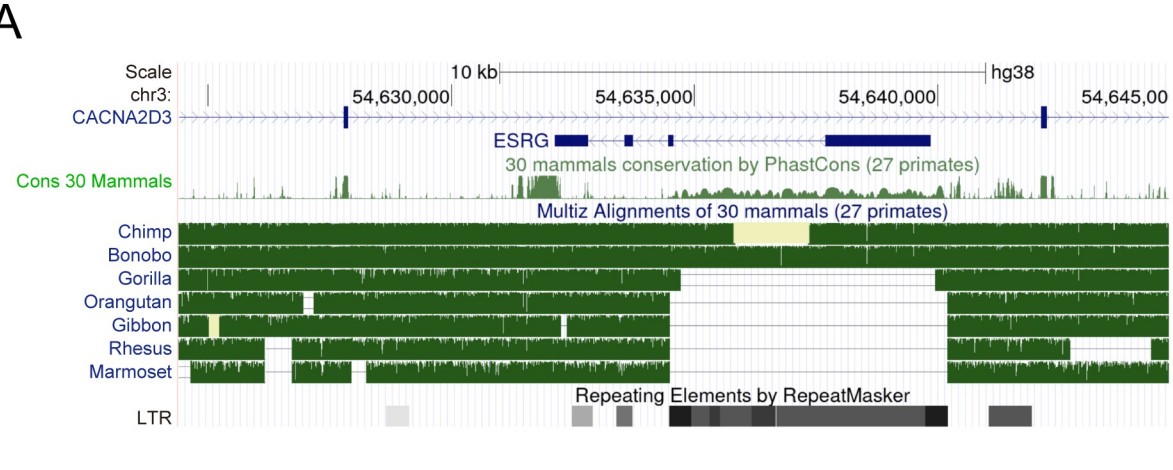

**Fig 1. Conservation analysis of ESRG.** (A) Modified screenshot from the UCSC genome browser showing the ESRG transcript in context to the RepeatMasker annotation, primate phastCons scores, and great ape and primate multiz-alignments. Note that the missing data in the chimpanzee were available in a newer chimpanzee assembly (panTro6) and was included in our later analysis. (B) DESeq2 normalized and variance stabilized expression in human and chimpanzee iPSCs and iPSC-derived cardiomyocytes (iPSC-CM). In iPSCs ESRG is similarly highly expressed as OCT3/4 and GAPDH, and completely downregulated in iPSC-CM. Moreover, in iPSCs ESRG is significantly higher expressed in humans than in chimpanzees ($\log_2$ fold change = 3.85; p-adj$<10^{-17}$; S2 Table). (C) Fraction of substitutions and SNVs across exons and introns of ESRG. Both diversity and divergence are highest in the LTR-region of exon 1. (D) Site frequency spectrum across 30,000 chromosomes across human populations for ESRG exons, other non-coding exons, synonymous and nonsynonymous sites of next gene CACNA2D3 and across the genome. (E) Distribution of the fraction of singletons for conserved lncRNAs (>5% sites with PhastCons>0.9) and other lncRNAs with at least 50 SNVs. Only very few have a singleton fraction that differs significantly from the neutral expectation as derived from synonymous sites ($\chi^2$-test; *p<0.05*, red tick-marks on the x-axis).

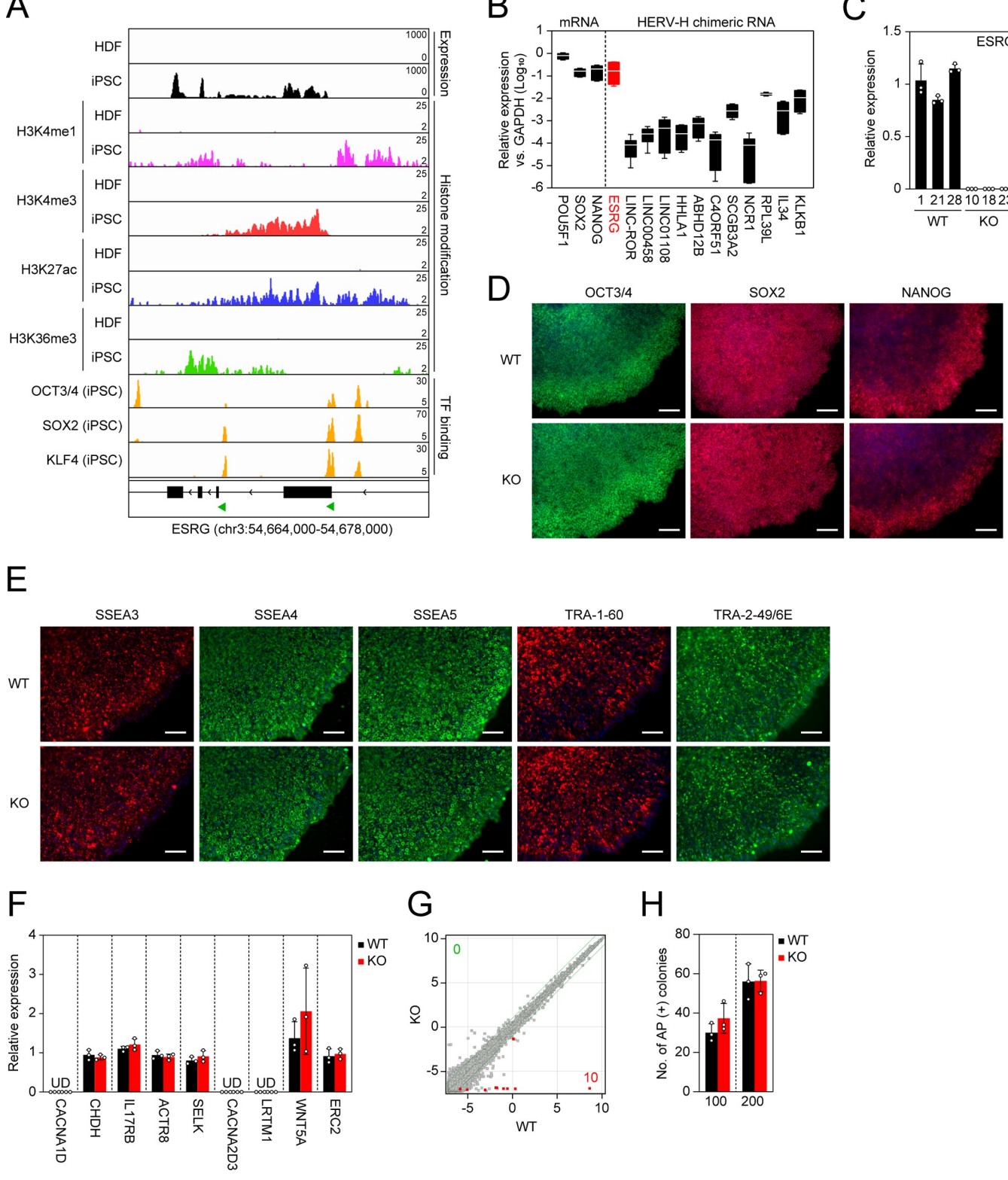

**Fig 2. ESRG is dispensable for primed pluripotency.** (A) Epigenetic status of the ESRG locus. We used the published RNA-seq (GSE56568) and ChIP-seq (GSE56567, GSE89976) data to confirm the RNA expression and the statuses of histone modifications and PSC core transcription factor (TF) binding on the ESRG locus in HDFs and iPSCs on human genome assembly hg19. The green arrowheads at the bottom indicate the location of the LTR7 elements. (B) Expression of PSC-associated mRNAs and HERV-H chimeric RNAs. Shown are the averaged expressions of the indicated transcripts in H9 ESCs, 585A1

iPSCs, and 201B7 iPSCs. Error bars and white lines indicate min. to max. and the mean of each gene expression, respectively. Values are compared to GAPDH. n = 3. (C) Expression of ESRG in ESRG WT and KO PSC clones. Values are normalized by GAPDH and compared with primed H9 ESCs. n = 3. (D) Expression of PSC core transcription factors. Bars,100 μm. (E) Expression of PSC-specific surface antigens. Bars, 100 μm. (F) Expression of neighbor genes <10 Mbp apart from ESRG gene. Values are normalized by GAPDH and compared with parental primed H9 ESCs. n = 3. (G) Global gene expression. Scatter plots compare the microarray data of ESRG WT and KO primed PSCs. The colored plots indicate differentially expressed genes (DEGs) with statistical significance (FC>2.0, FDR, 0.05). The numbers of DEGs (FC>2.0, FDR,0.05) are shown in the figure. n = 3. (H) Plating efficiency. Shown are the number of AP (+) colonies raised from 100 or 200 ESRG WT and KO PSCs. n = 3. Numerical values for B, C, F, and H are available in S1 Data.

fetal tissues (S1A Fig). Compared to other PSC-associated HERV-H chimeric transcripts, ESRG expression exhibits a sharp contrast between human PSCs and somatic tissues [8,10,15–17]. Furthermore, ESRG is expressed in human PSCs, including embryonic carcinoma cell (ECC) lines, but is silenced in four cancer cell lines and ten cell lines derived from normal tissues (S1B Fig). Quantitative reverse transcription-polymerase chain reaction (qRT-PCR) revealed that the ESRG expression is significantly higher than the expression of other HERV-H-related transcripts and is comparable to the expression of SOX2 and NANOG, which play essential roles in pluripotency, in three independent human PSC lines (Fig 2B). These data suggest that ESRG expression is abundant in human PSCs and is tightly silenced in differentiated states.

## ESRG is dispensable for human pluripotency

The above results showing low conservation but high expression in humans led us to test the function of ESRG in human PSCs. To make a complete loss of function of the lncRNA ESRG, we employed a CRISPR/Cas9 platform and two small guide RNAs (sgRNAs) to delete ~8,400 bp of the genomic region including the entire ESRG gene (Figs 2A and S2A). As a result, we obtained multiple independent ESRG knockout (KO) PSC lines that exhibit complete deletion of the gene body with unique minor deletion patterns in both alleles under a primed PSC culture condition (S2B and S2C Fig). In this study, we used three clones as wild-type (WT) controls carrying intact ESRG alleles with no or minor deletions at the sgRNA recognition sites (S2D Fig). The expression of ESRG was undetectable in the KO clones by qRT-PCR (Fig 2C). Immunocytochemistry showed that ESRG KO PSCs express the PSC core transcription factors (Fig 2D) and PSC-specific surface antigens (Fig 2E). The loss of ESRG made no impact on the expression of neighbor genes located within 10 Mbp of ESRG (Fig 2F). Global transcriptome analysis by microarray revealed that the loss of ESRG altered the expression of only six genes (10 probes in microarray) such as ESRG (Chr. 3), TMLHE (Chr. X), LDHC (Chr. 11), LOC339975 (Chr. 4), AIFM2 (Chr. 10), XLOC_L2_01411 (Chr. 4) and lnc-CDKAL1-1 (Chr. 6) between ESRG WT and KO PSCs in primed condition (Fig 2G). We also confirmed that loss of ESRG affects the expression of 36 genes which are located widely on different chromosomes by RNA-seq (S3 Fig). Only THELE, LDHC, and ESRG itself were found as differentially expressed genes (DEGs) common in microarray and RNA-seq data. These data suggest that ESRG has no apparent cis-acting lncRNA function by interacting with neighbor genes. Moreover, ESRG KO PSCs normally survived while maintaining the undifferentiated state as judged by alkaline phosphatase (AP) activity and the absence of any apparent genomic abnormalities (Figs 2H and S4). Altogether, these data suggest that loss of ESRG does not affect the self-renewal of human primed PSCs.

We revisited the shRNA-mediated KD of ESRG to confirm the consistency with the phenotype of ESRG loss. Three independent shRNAs [8,9] decreased the ESRG expression to 16.38~32.55% compared to the parental line (S5A Fig). After 20 days of shRNA transduction, the RNA expression of POU5F1 and/or NANOG were reduced by two of three shRNAs

(shESRG-4 and 5), although the most effective shRNA (shESRG-2) against ESRG did not alter them (S5A Fig). None of ESRG shRNAs induced the expression of early differentiation markers such as T (mesendoderm) and NES (neuroectoderm) (S5A Fig). The ESRG KD PSCs grew normally with expressing NANOG protein (S5B Fig). These data suggest that ESRG KD by shRNAs does not induce the differentiation of human PSCs in the primed state. We and others previously reported the effects of shRNA-mediated pan HERV-H KD on human PSC characteristics [8–10]. Three shRNAs against the conserved regions of HERV-Hs decreased to 29.06~56.48% compared to the parental line (S6A Fig). One of them (shHERVH-1), as similar efficiency of the ESRG shRNAs, finely knocked down the ESRG expression to 14.55% of the parental line (S5B and S6B Figs). Microarray data suggested that no noticeable changes were detected in the expression of PSC markers and lineage markers (S6B Fig). In addition to the transcriptome data, we confirmed that all three HERV-H KD PSC lines were able to expand with maintaining the stem cell morphologies and NANOG protein expression (S6C Fig). These data support that ESRG is dispensable for the self-renewing of primed PSCs.

In addition to the primed state, we tested if ESRG is required for another state of pluripotency, the so-called naïve state, which also expresses ESRG but at a significantly lower level than the primed state (Fig 3A). Regardless of the ESRG expression, naïve PSCs could be established by switching the media composition and could self-renew while keeping a tightly packed colony formation (Fig 3B) [29–31]. Furthermore, they exhibited a significantly high expression of the naïve pluripotency markers KLF4 and KLF17 and attenuated the expression of the primed PSC marker ZIC2 (Fig 3C) [32,33]. Twenty-nine genes including ESRG and CNCNA2D3 were found as DEGs between ESRG WT and KO PSCs in naïve condition by RNA-seq (S3 Fig), although microarray analysis revealed that ESRG had no effect on the global gene expression of naïve PSCs (Fig 3D). Altogether, these data suggest that ESRG does not contribute to self-renewal and gene expression of human naïve PSCs.

We also differentiated ESRG WT and KO naïve PSCs to the primed pluripotent state. As a result, irrespective of the ESRG genotype, we detected the hallmarks of primed pluripotency such as flatter colony formation, the reactivation of ZIC2 and the suppression of KLF4 and KLF17, suggesting the bidirectional transition between naïve and primed pluripotency does not require ESRG (Fig 3E and 3F). Taken together, these data demonstrate that ESRG is dispensable for the maintenance of human PSCs.

## ESRG is not involved in differentiation

Next, we analyzed whether ESRG is required for the differentiation of human primed PSCs by embryoid body (EB) formation. The absence of ESRG had no effect on EB formation by floating culture or differentiation into trilineage such as alpha-fetoprotein (AFP) positive (+) endoderm, smooth muscle actin (SMA) (+) mesoderm, and βIII-TUBULIN (+) ectoderm (Fig 4A and 4B). Other lineage markers such as DCN (endoderm), MSX1 (mesoderm) and MAP2 (ectoderm) were also well induced in EBs derived from either ESRG WT or KO primed PSCs (Fig 4C). Global transcriptome analysis by microarray indicated the loss of ESRG caused no significant gene expression changes during EB differentiation (Fig 4D). These data suggest that ESRG KO PSCs retained the potential to differentiate into all three germ layers.

Previous studies showed that HERV-H expression regulates the neural differentiation potential of human PSCs [10,15,34]. Thus, in addition to the random differentiation by EB formation, we tested whether ESRG contributes to the directed differentiation of human primed PSCs into NPCs by the dual SMAD inhibition method [35,36]. Both ESRG WT and KO PSCs were able to differentiate into expandable NPCs, which expressed the early neural lineage marker PAX6 but not OCT3/4 (Fig 4E). Other NPC markers such as SOX1 and NES were well

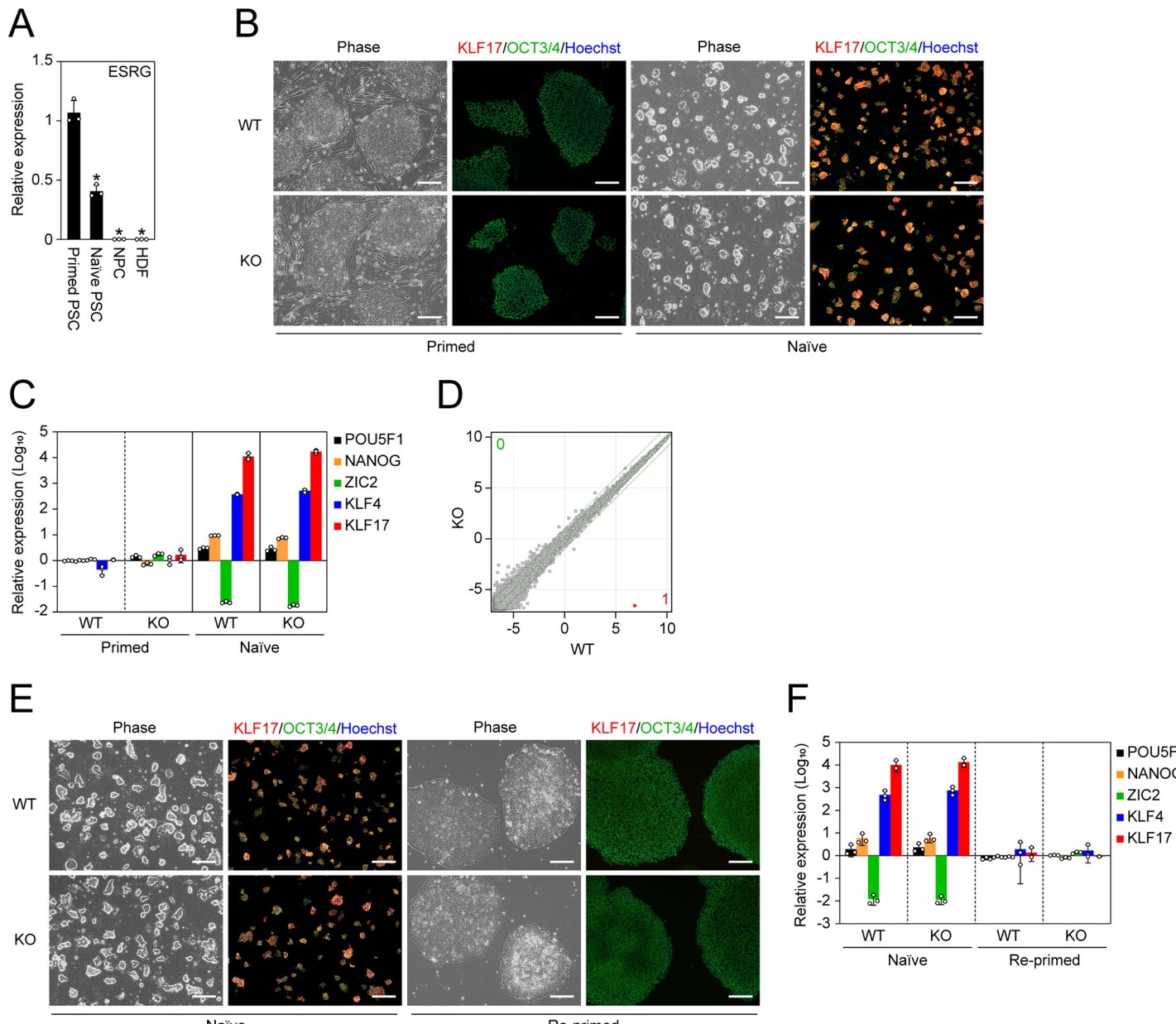

**Fig 3. No impact of ESRG on naïve pluripotency.** (A) The ESRG expression. Shown are relative expressions of ESRG in primed PSCs, naïve PSCs, NPCs and HDFs. Values are normalized by GAPDH and compared with the primed 585A1 iPSC line. *P<0.05 vs. primed PSCs by unpaired t-test. n = 3. (B) Conversion to naïve pluripotency. Shown are representative images of ESRG WT and KO primed and naïve PSCs under phase contrast and of immunocytochemistry for KLF17 (red) and OCT3/4 (green). Bars, 200 μm. (C) The expression of primed and naïve PSC markers. Shown are the relative expressions of common PSC markers (POU5F1 and NANOG), a primed PSC marker (ZIC2) and naïve PSC markers (KLF4 and KLF17). Values are normalized by GAPDH and compared with primed H9 ESCs. n = 3. (D) Global transcriptome. Scatter plots comparing the microarray data of ESRG WT and KO naïve PSCs. The colored plot indicates DEG with statistical significance (FC>2.0, FDR,0.05). The numbers of DEGs (FC>2.0, FDR,0.05) are shown in the figure. n = 3. (E) Differentiation to primed pluripotency. Representative images of ESRG WT and KO naïve PSCs before and after conversion to the primed pluripotent state are shown. Bars, 200 μm. (F) The expression of primed and naïve PSC markers. Shown are the relative expressions of the marker genes in (C) in ESRG WT and KO naïve PSCs before and after the differentiation to the primed pluripotent state. Values are normalized by GAPDH and compared with primed H9 ESCs. n = 3. Numerical values for A, C, and F are available in S1 Data.

induced, whereas the PSC marker NANOG was silenced (Fig 4F). These data suggest that ESRG is not responsible for HERV-H-regulated neural differentiation. Taken together, we concluded that ESRG is not required for the differentiation of human PSCs.

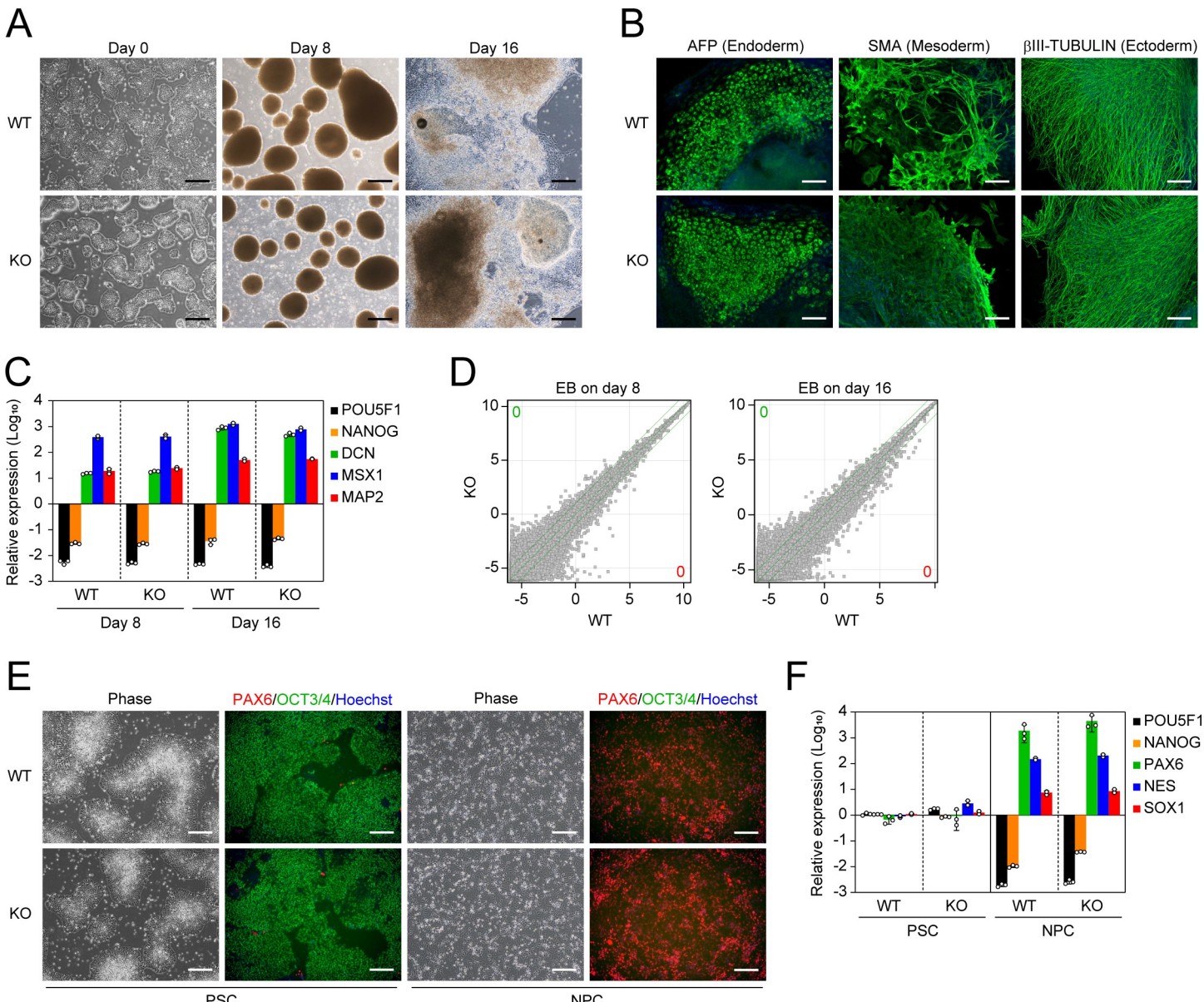

**Fig 4. ESRG-deficient PSCs are capable of differentiating.** (A) Differentiation by EB formation. Bars, 500 μm. (B) Trilineage differentiation. Bars, 200 μm. (C) The expression of differentiation markers. Shown are the relative expressions of PSC markers (POU5F1 and NANOG) and differentiation markers (DCN, MSX1, and MAP2) on days 8 and 16 of EB differentiation. Values are normalized by GAPDH and compared with primed H9 ESCs. n = 3. (D) Global gene expression of differentiation derivatives. Scatter plots compare the microarray data of ESRG WT and KO PSC-derived EBs on days 8 and 16. The numbers of DEGs (FC>2.0, FDR,0.05) are shown in the figure. n = 3. (E) NPC differentiation. Representative images of ESRG WT and KO PSCs and NPCs under phase contrast and of immunocytochemistry for PAX6 (red) and OCT3/4 (green) are shown. Bars, 200 μm. (F) The expression of NSC markers. Shown are the relative expressions of PSC markers (POU5F1 and NANOG) and NPC markers (PAX6, SOX1, and NES) in ESRG WT and KO PSCs and NPCs. Values are normalized by GAPDH and compared with primed H9 ESCs. n = 3. Numerical values for C and F are available in S1 Data.

## ESRG is not required for somatic cell reprogramming toward pluripotency

A previous study showed that the overexpression of ESRG improves iPSC generation [8], suggesting a positive effect on somatic cell reprogramming toward pluripotency. The activation of ESRG in the early stage of reprogramming and the high expression of ESRG during reprogramming support this hypothesis (Fig 5A) [20]. Therefore, we reprogrammed ESRG WT and KO NPCs to iPSCs by introducing OSK. iPSCs emerged from ESRG WT and KO NPCs with

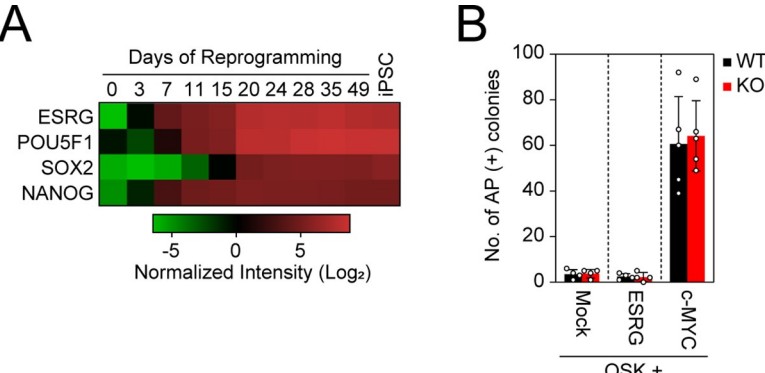

**Fig 5. ESRG is dispensable for iPSC reprogramming.** (A) The expression of ESRG during reprogramming. The heatmap generated by using the dataset (GSE54848) shows the normalized intensities of ESRG, POU5F1 (endogenous), SOX2 (endogenous), and NANOG expression from microarray data in the time course of iPSC reprogramming (days 0–49) and established iPSCs (far right). n = 3. (B) The effect of ESRG on iPSC generation. Shown are the numbers of AP (+) iPSC colonies 24 days after the transduction of OSK along with Mock (n = 4), ESRG (n = 4), and c-MYC (n = 5). Numerical values for A and B are available in S1 Data.

comparable efficiency (Fig 5B). This observation suggests that ESRG is dispensable for iPSC generation. In addition, along with OSK, we transduced c-MYC, a potent enhancer of iPSC generation [37,38], or exogenous ESRG. c-MYC but not exogenous ESRG increased the efficiency of the iPSC generation from ESRG WT and KO NPCs equally (Fig 5B). Taken together, these data suggest that ESRG has no impact on somatic cell reprogramming toward iPSCs.

## Discussion

In this study, we completely excised the entire ESRG gene to understand its role in human PSCs while avoiding residual expression and off-target effects. As a result, ESRG KO PSCs showed no apparent phenotypes in self-renewal and differentiation potential. A previous study showed the importance of ESRG in human PSC identity by using an shRNA-mediated KD approach [8]. Although we used the same H9 ESC line as that study, the different strategies for the loss of function and subsequent experiments, such as KD and KO, may explain the different results. Therefore, this study revisited the ESRG KD by using three shRNAs including published sequences [8]. Indeed, two published shRNAs (shESRG-4 and 5) decreased POU5F1 (84.28 and 55.28% of the parental line) and NANOG (52.66 and 67.14% of the parental line), respectively, whereas shESRG-2 that is newly designed in this study did not change their expression (103.54 (POU5F1) and 106.64% (NANOG) of the parental line) (S5A Fig). The reduction of PSC marker expression that varied among shRNAs was not enough to induce the differentiation of human PSCs (S5C Fig). In addition to the ESRG KD, we also showed the effects of pan HERV-H KD in human PSCs in primed condition (S6 Fig). We previously showed that the suppression of HERV-H expression using shRNA did not disrupt the self-renewal of human PSCs [10,34]. A recent paper by Zhang et al. showed that pan-HERV-H KD in human PSCs by using CRISPR interference did not induce spontaneous differentiation like we observed [39]. However, since other groups concluded that HERV-H KD induced differentiation [8,9], further studies are required to understand what HERV-H is doing. One possibility that may explain the discrepancy of the results between previous and current studies [8] is the off-target effect of RNAi. Similar observations have been found for the role of lncRNA Cyrano that is highly conserved in mice and humans. Knockdown by using shRNA suggested Cyrano lncRNA maintains mouse PSC identity [40], but targeted deletion of the Cyrano gene and gene silencing by CRISPR interference demonstrated no impact on

the mouse or human PSC identity [41–43]. Further, it has been argued that the shRNA-mediated KD of nuclear lncRNAs might be difficult or inefficient compared to cytoplasmic RNAs such as mRNAs [44,45]. In addition, while small nucleotide insertions or deletions causing frameshift of the reading frames work well for the loss of function of protein-coding genes, the same is not true for non-coding RNAs. In this context, our study succeeded in generating the complete deletion of ESRG gene alleles, providing highly reliable results.

This study clearly demonstrated that ESRG is dispensable for human PSC identity. Neither primed nor naïve PSCs require ESRG for their identities, such as colony morphology or gene expression signatures, meaning ESRG is dispensable for human pluripotency, at least in an in vitro culture environment. However, since ESRG is expressed in epiblast-stage human embryos [8,46], it might be involved in early human embryogenesis.

ESRG is stochastically activated by OSK in rare reprogrammed intermediates that have the potential to become bona fide iPSCs and is highly expressed throughout the process of reprogramming toward iPSCs [20]. In the present study, we showed that ESRG KO NPCs can be reprogrammed with the same efficiency as ESRG WT NPCs. These data suggest that ESRG is a good marker of the intermediate cells in the early stage of reprogramming rather than a functional molecule that is needed for iPSC generation.

In summary, this study provides clear evidence of the dispensability of ESRG for human PSC identities, such as global gene expressions and differentiation potentials, in two distinct types of pluripotent states. We also demonstrated that the function of ESRG is not required for recapturing pluripotency via somatic cell reprogramming. Finally, the tightly regulated and high expression of ESRG promises to make an excellent marker of undifferentiated human PSCs both in basic research and clinical application [20,47].

## Methods

### Expression conservation

To investigate ESRG expression, we used an RNA-seq data set that investigated cardiomyocyte differentiation from human and chimpanzee iPSCs [24]. Read count matrices were downloaded from Gene Expression Omnibus (GSE110471). We selected iPSC and iPSC-derived cardiomyocyte samples and filtered the data for genes that were detected in at least 40% of the samples and had an average expression of at least 5 counts, yielding a final matrix with 17,213 genes. Differential expression analyses and variance-stabilizing transformation were performed using DESeq2 v.1.30.0 [48], using a model including the factors ~cell type: species + species. iPSC-specific differential expression between human and chimpanzee was inferred via the interaction term identifying iPSC-specific differences between human and chimpanzee.

### Multiple sequence alignment

We used the human ESRG sequence (+20 kb in each direction) (NCBI 105.20190906 Reference Sequence NR_027122.1; hg19) to search orthologous sequence in the great apes genomes: chimpanzee (Pan troglodytes, GCF_002880755.1), bonobo (Pan paniscus, GCF_013052645.1), gorilla (Gorilla gorilla, GCA_900006655.3) and orang (Pongo abelii, GCF_002880775.1) using dc-megablast with default options [49]. Finally, the identified regions were aligned into a multiple sequence alignment using mafft [50] and manual inspection.

### Human polymorphism data

We identified the polymorphic sites based on gnomAD v2.1.1 database [28]. We downloaded the vcf-file and tsv coverage files derived from whole-genome sequencing of 15,708 unrelated

individuals. For further analyses, we only used bi-allelic single nucleotide variants (SNVs) that also passed the quality criteria of gnomAD and had at least 15x coverage in at least 95% of the individuals. To balance small differences in the numbers of chromosomes sampled at each polymorphic site, we downsampled it to 30,000. In the following, we analyze synonymous and non-synonymous SNVs and SNVs falling into the exons of long non-coding RNAs (Gencode version 35, transcript type 'lncRNA', lifted over to hg19 using hg38ToHg19 UCSC chain file [51]). For ESRG, we distinguish SNPs falling into exons, introns, and LTR-derived sequences and compare them to the surrounding protein-coding gene CACNA2D3.

## The culture of primed PSCs

H9 ESC (RID:CVCL_9773) [52] and 585A1 iPSC (RRID:CVCL_DQ06) [53] lines were maintained in StemFiT AK02 media (Ajinomoto) supplemented with 100 ng/ml recombinant human basic fibroblast growth factor (bFGF, Peprotech) (hereafter F/A media) on a tissue culture plate coated with Laminin 511 E8 fragment (LN511E8, NIPPI) [54,55]. N18 iPSC line was maintained in F/A media supplemented with 1 μg/ml of doxycycline on a tissue culture plate coated with LN511E8 [34]. 201B7 iPSC (RRID:CVCL_A324) line was cultured on mitomycin C (MMC)-inactivated SNL mouse feeder cells (RRID:CVCL_K227) in Primate ESC Culture medium (ReproCELL) supplemented with 4 ng/ml bFGF [12].

## Induction and maintenance of naïve PSCs

The conversion of primed PSCs to the naïve state was performed as described previously [31]. Prior to naïve conversion, primed PSCs were maintained on MMC-treated primary mouse embryonic fibroblasts (PMEFs) in DFK20 media consisting of DMEM/F12 (Thermo Fisher Scientific), 20% Knockout Serum Replacement (KSR, Thermo Fisher Scientific), 1% MEM non-essential amino acids (NEAA, Thermo Fisher Scientific), 1% GlutaMax (Thermo Fisher Scientific) and 0.1 mM 2-mercaptoethanol (2-ME, Thermo Fisher Scientific)) supplemented with 4 ng/ml bFGF. The cells were harvested using CTK solution (ReproCELL) and dissociated into single cells. One hundred thousand cells were plated onto MMC-treated PMEFs in a well of a 6-well plate in DFK20 media plus bFGF and 10 μM Y-27632. Thereafter, the cells were incubated in hypoxic condition (5% $O_2$). On the next day, the media was replaced with NDiff227 (Takara) supplemented with 1 μM PD325901 (Stemgent), 10 ng/ml of recombinant human leukemia inhibitory factor (LIF, EMD Millipore), and 1 mM Valproic acid (Wako). Three days later, the media was switched to PXGL media (NDiff227 supplemented with 1 μM PD325901, 2 μM XAV939 (Wako), 2 μM Gö6983 (Sigma Aldrich), and 10 ng/ml of LIF). When round shape colonies were visible (around day 9 of the conversion), the cells were dissociated using TrypLE Express (Thermo Fisher Scientific) and plated onto a new PMEF feeder plate in PXGL media plus 10 μM Y-27632. The media was changed daily, and the cells were passaged every 4–5 days. Cells after at least 30 days of the conversion were used for the assays.

## Differentiation of naïve PSCs to the primed state

Naïve PSCs were harvested using TrypLE Express and plated at 5 x $10^5$ cells onto a well of a LN511E8-coated 6-well plate in PXGL media supplemented with 10 μM Y-27632. On the next day, the media was replaced with F/A media. After 2 and 8 days, the cells were harvested and split to a new LN511E8-coated plate in F/A media plus 10 μM Y-27632. On day 16 of the differentiation, the cells were fixed for immunocytochemistry, and RNA samples were collected to analyze the marker gene expression.

## Induction and maintenance of NPCs

Primed PSCs were differentiated into expandable NPCs by using the STEMdiff SMADi Neural Induction Kit (Stem Cell Technologies) as previously described [34–36]. In brief, primed PSCs were maintained on a Matrigel (Corning)-coated plate in mTeSR1 media (Stem Cell Technologies) prior to the NPC induction. The cells were harvested using Accutase (EMD Millipore) and transferred at $3 \times 10^6$ cells to a well of an AgrreWell800 plate (Stem Cell Technologies) in STEMdiff Neural Induction Medium + SMADi (Stem Cell Technologies) supplemented with 10 μM Y-27632. Five days later, uniformly sized aggregates were collected using a 37 μm Reversible Strainer (Stem Cell Technologies) and plated onto a Matrigel-coated 6-well plate in STEMdiff Neural Induction Medium + SMADi. Seven days later, neural rosette structures were selectively removed by using STEMdiff Neural Rosette Selection Reagent (Stem Cell Technologies) and plated onto a new Matrigel-coated 6-well plate in STEMdiff Neural Induction Medium + SMADi. After that, the cells were passaged every 2–3 days until day 30 post-differentiation. The established NPCs were maintained on a Matrigel-coated plate in STEMdiff Neural Progenitor Medium (Stem Cell Technologies) and passaged every 3–4 days.

## The culture of other cells

HDFs and PLAT-GP packaging cells (RRID:CVCL_B490) were cultured in DMEM (Thermo Fisher Scientific) containing 10% fetal bovine serum (FBS, Thermo Fisher Scientific).

## Embryoid body (EB) differentiation

PSCs were cultured on a Matrigel-coated plate in mTeSR1 media until reaching confluency prior to EB formation. The cells were harvested using CTK solution (ReproCELL), and cell clumps were transferred onto an ultra-low binding plate (Corning) in DFK20 media. For the first 2 days, 10 μM Y-27362 was added to the media to improve cell survival. The media was changed every other day. After 8 days of floating culture, the EBs were transferred onto a tissue culture plate coated with 0.1% gelatin (EMD Millipore) and maintained in DFK20 media for another 8 days.

## Plasmid

Full-length ESRG complementary DNA (cDNA) was amplified using ESRG-S and ESRG-AS primers and inserted into the BamHI/NotI site of a pMXs retroviral vector [56] using In-Fusion technology (Clontech). The primer sequences for the cloning are available in S5 Table. For the KD experiments, we used transposon vectors such as Sleeping Beauty (SB) and Piggy-Bac (PB) that contain mouse U6 promoter, drug selection markers and the genes encoding fluorescent proteins [34]. The shRNA sequences are provided in S5 Table.

## Reprogramming

Retroviral transduction of the reprogramming factors was performed as described previously [12,20]. A pMXs retroviral vector encoding human OCT3/4 (RRID:Addgene_17217), human SOX2 (RRID:Addgene_17218), human KLF4 (RRID:Addgene_17219), human c-MYC (RRID: Addgene_17220) and ESRG (6 μg each) along with 3 μg of pMD2.G (gift from Dr. D. Trono; RRID:Addgene_12259) was transfected into PLAT-GP packaging cells, which were plated at $3.6 \times 10^6$ cells per 100 mm dish the day before transfection, using FuGENE6 transfection reagent (Promega). Two days after the transfection, virus-containing supernatant was collected and filtered through a 0.45 μm-pore size cellulose acetate filter to remove the cell debris. Viral particles were precipitated using Retro-X Concentrator (Clontech) and resuspended in

STEMdiff Neural Progenitor Medium containing 8 μg/ml Polybrene (EMD Millipore). Then, appropriate combinations of viruses were mixed and used for the transduction to NPCs. This point was designated day 0. The cells were harvested on day 3 post-transduction and replated at $5 \times 10^4$ cells per well of a LN511E8-coated 6-well plate in STEMdiff Neural Progenitor Medium. The following day (day 4), the medium was replaced with F/A media, and the medium was changed every other day. The iPSC colonies were counted on day 24 post-transduction. Bona fide iPSC colonies were distinguished from non-iPSC colonies by their morphological differences and/or alkaline phosphatase activity.

## Deletion of ESRG gene

Two days before a ribonucleoprotein (RNP) complex transfection, we introduced a small interfering RNA (siRNA) against TP53 gene (s605, Thermo Fisher Scientific) to H9 ESCs (passage number 49) using Lipofectamine RNAi Max (Thermo Fisher Scientific) according to the manufacturer's protocol [57,58]. An RNP complex consisting of 40 pmol of Alt-R S.p. HiFi Cas9 Nuclease V3 (Integrated DNA Technologies) and two single guide RNAs (sgRNAs: sgESRG-U (5'-AGAGAAUACGAAGCUAAGUG-3') and sgESRG-L (5'-AUUGCAGUU GUCACAUGACA-3'), 150 pmol each; SYNTHEGO) was introduced into $5 \times 10^5$ of siRNA-transfected cells using a 4D-Nucleofector System with X Unit (Lonza) and P3 Primary Cell 4D-Nucleofector Kit S (Lonza) with the CA173 program. Three days after the nucleofection, the cells were harvested and replated at 500 cells onto a LN511E8-coated 100 mm dish in F/A media supplemented with 10 μM Y-27632. The cells were maintained until the colonies grew big enough for subcloning. The colonies were mechanically picked up, dissociated using TrypLE select, and plated onto a LN511E8-coated 12-well plate in F/A media supplemented with 10 μM Y-27632.

The genomic DNA of the expanded clones was purified using the DNeasy Blood & Tissue Kit (QIAGEN). Fifty nanograms of purified DNA was used for quantitative polymerase chain reaction (PCR) using TaqMan Genotyping Master Mix (Thermo Fisher Scientific) on an ABI7900HT Real Time PCR System (Applied Biosystems). TaqMan Assays (Thermo Fisher Scientific) such as ESRG_cn1 (Hs05898393_cn) and ESRG_cn2 (Hs06675423_cn) detected the ESRG locus and TaqMan Copy Number Reference Assay human RNase P (4403326, Thermo Fisher Scientific) was used as an endogenous control. To verify the indel patterns in wild-type clones, fragments around the sgESRG-U and sgESRG-L recognition sites were amplified with ESRG-U-S/ESRG-U-AS and ESRG-L-S/ESRG-L-AS primer sets, respectively. The amplicons were purified using the QIAquick PCR Purification Kit (QIAGEN) and subjected to sequencing. To check the deleted sequences in the knockout clones, a fragment with ESRG-U-S/ESRG-L-AS primers was amplified. Conventional PCR was performed using KOD Xtreme Hot Start DNA Polymerase (EMD Millipore). The fragments were cloned into pCR-Blunt II TOPO using the Zero Blunt TOPO PCR Cloning Kit (Thermo Fisher Scientific), and the sequencing was verified using M13 forward and M13 reverse universal primers. The sequence data was analyzed using SnapGene software (GSL Biotech LLC). The primer sequences are provided in S5 Table.

## RNA isolation and reverse-transcription polymerase chain reaction

The cells were lysed with QIAzol reagent (QIAGEN), and the total RNA was purified using a miRNeasy Mini Kit (QIAGEN) according to the manufacturer's protocol. The reverse transcription (RT) of 1 μg of purified RNA was done by using SuperScript III First-Strand Synthesis SuperMix (Thermo Fisher Scientific). Quantitative RT-PCR was performed using TaqMan Assays with TaqMan Universal Master Mix II, no UNG (Applied Biosystems) or using gene-

specific primers with THUNDERBIRD Next SYBR qPCR Mix (TOYOBO) on an ABI7900HT or a QuantoStudio 5 Real Time PCR System (Applied Biosystems). The $C_t$ values of the undetermined signals caused by too low expression were set at 40. The levels of mRNA were normalized to the ACTB or GAPDH expression, and the relative expression was calculated as the fold-change from the control. Information about the primers and TaqMan Assays are shown in S5 and S6 Tables, respectively.

## Gene expression analysis by microarray

The total RNA samples were purified using the miRNeasy Mini Kit, and the quality was evaluated using a 2100 Bioanalyzer (Agilent Technologies). Two hundred nanograms of total RNA was labeled with Cyanine 3-CTP and used for hybridization with SurePrint G3 Human GE 8x60K (version 1 (G4851A) and version 3 (G4851C), Agilent Technologies) and the one-color protocol. The hybridized arrays were scanned with a Microarray Scanner System (G2565BA, Agilent Technologies), and the extracted signals were analyzed using the GeneSpring version 14.6 software program (Agilent Technologies). Gene expression values were normalized by 75th percentile shifts. Differentially expressed genes between ESRG WT and KO ESCs were extracted by t-tests with Benjamini and Hochberg corrections [fold change (FC) > 2.0, false-discovery rate (FDR) < 0.05].

## RNA sequencing (RNA-seq) and data analysis

Total RNAs were extracted and purified using the miRNeasy Mini kit and RNase-Free DNase Set (QIAGEN) according to the manufacturer's manuals. Libraries were constructed by Tru-Seq Stranded total RNA with the Ribo-Zero Gold LT Sample Prep Kit, Set A and B (Illumina), according to the manufacturer's manual. For sequencing by using NovaSeq 6000, the NovaSeq 6000 S1 Reagent Kit v1.5 (100 cycle) (Illumina) was used. We trimmed adapter sequences by using cutadapt-1.18 [59], removed the reads mapped to ribosomal RNA by using bowtie2 (version 2.2.5) and samtools (version 1.7) [60,61], mapped the reads to the human genome (hg38 from the UCSC Genome Browser) by using STAR (version 2.5.3a) [62], conducted a quality check by using RSeQC (version 2.6.4) [63], counted the reads by using HTSeq (version 0.11.2) with the GENCODE annotation file (version 27) [64,65], and normalized the counts by using DESeq2 (version 1.24.0) in R (version 3.6.1) [48]. Using the DESeq2 package, Wald tests were performed.

## Immunocytochemistry

The cells were washed once with PBS, fixed with fixation buffer (BioLegend) for 15 min at room temperature and blocked in PBS containing 1% bovine serum albumin (BSA, Thermo Fisher Scientific) and 2% normal donkey serum (Sigma-Aldrich) for 45 min at room temperature. For the staining of intracellular proteins, the fixed cells were permeabilized by adding 0.2% TritonX-100 (Teknova) during the blocking process. Then the cells were incubated with primary antibodies diluted in PBS containing 1% BSA at 4˚C overnight. After washing with PBS, the cells were incubated with secondary antibodies diluted in PBS containing 1% BSA and 1 µg/ml Hoechst 33342 (Thermo Fisher Scientific) for 45 min at room temperature in the dark. The fluorescent signals were detected using a BZ-X710 imaging system (KEYENCE). The antibodies and dilution rate were as follows: anti-OCT3/4 (1:250, 611203, BD Biosciences), anti-SOX2 (1:100, ab97959, Abcam), anti-NANOG (1:100, ab21624, Abcam), anti-KLF17 (1:100, HPA024629, Atlas Antibodies), anti-PAX6 (1:1,000, 901301, BioLegend), SSEA3 (1:100, 09–0044, Stemgent), SSEA4 (1:100, 09–0006, Stemgent), SSEA5 (1:100, 355201, BioLegend), TRA-1-60 (1:100, MAB4360, EMD Millipore), TRA-2-49/6E (1:100, 358702,

BioLegend), anti-AFP (1:200, GTX15650, GeneTex), anti-SMA (1:200, CBL171-I, EMD Millipore), anti-βIII-TUBULIN (1:1,000, XMAB1637, EMD Millipore), Alexa 488 Plus anti-mouse IgG (1:500, A32766, Thermo Fisher Scientific), Alexa 647 Plus anti-mouse IgG (1:500, A32787, Thermo Fisher Scientific), Alexa 647 Plus anti-rabbit IgG (1:500, A32795, Thermo Fisher Scientific), Alexa 594 anti-rat IgM (1:500, A21213, Thermo Fisher Scientific) and Alexa 555 anti-mouse IgM (1:500, A21426, Thermo Fisher Scientific).

### Quantification and statistical analysis

Data are presented as the mean ± standard deviation unless otherwise noted. Sample number (n) indicates the number of replicates in each experiment. The number of experimental repeats is indicated in the figure legends. To determine statistical significance, we used the unpaired t-test for comparisons between two groups using Excel Microsoft 365 (Microsoft). Statistical significance was set at $p < 0.05$. Graphs and heatmaps were generated using GraphPad Prism 8 software (GraphPad).

### Supporting information

**S1 Fig. ESRG expression profiles.** Expression of ESRG in human tissues. (A) Shown are the normalized intensities of ESRG expression from the microarray data of PSC (H9 ESC), 24 human adult tissues, and five fetal tissues. (B) Expression of ESRG in human cell lines. The normalized intensities of ESRG expression from the microarray data of several PSC lines including H9 ESC, 201B7 iPSC, 585A1 iPSC, 2102Ep embryonic carcinoma cells (ECC) and NTERA-2 ECC, cancer cell lines such as MCF7, HepG2, HeLa and Jurkat, and normal tissue-derived cells such as adipose tissue-derived mesenchymal stem cells (AdMSC), dental pulp-derived MSCs (DpMSC), human dermal fibroblasts (HDF), peripheral blood mononuclear cells (PBMC), bronchial epithelial cells (BrEC), prostate epithelial cells (PrEC), hepatocytes (Hep), epidermal keratinocytes (EKc), neural progenitor cells (NPC) and astrocytes (Astrocyte) are shown. Numerical values for A and B are available in S1 Data.
(TIF)

**S2 Fig. Deletion of ESRG locus.** (A) The scheme of ESRG targeting. The locations of sgRNAs for targeting (sgESRG-U and -L), primers for genotyping (U-S/AS and L-S/AS) and TaqMan Assays for copy number analyses (cn1 and cn2) are shown. The sequences of sgRNAs and primers are provided in the Methods section and S5 Table. (B) The copy number of the ESRG gene. The copy number of ESRG gene in ESRG WT (clones 1, 21 and, 28), a heterozygous clone (Het) that lacks one ESRG allele and KO (clones 10, 18 and, 23) were quantified by qPCR using TaqMan Copy Number Assays (cn1 and 2). Values are normalized by RNase P and compared with parental H9 ESCs. n = 3. (C) The sequences around the deletion sites in ESRG KO ESC clones verified by Sanger sequencing. (D) The sequences around the sgRNA recognition sites upstream (sgESRG-U) and downstream (sgESRG-L) of the ESRG locus in ESRG WT ESC clones verified by Sanger sequencing. Numerical values for B are available in S1 Data.
(TIF)

**S3 Fig. Validation of microarray results with RNA sequencing.** Global gene expression. Scatter plots compare $\log_2$ (Normalized count) of the RNA-seq data of ESRG WT and KO primed (left and naïve (right) PSCs. The colored plots indicate differentially expressed genes (DEGs) with statistical significance (FC>2.0, adjusted p-value <0.05). Three clones of ESRG WT and KO PSCs at different three passage numbers were analyzed in each condition.
(TIF)

**S4 Fig. Karyotypes of PSC clones used in the study.** Representative images of G-band staining show that all clones used in the study maintained normal female karyotypes (46XX). (TIF)

**S5 Fig. Knockdown of ESRG did not induce differentiation of human PSCs.** (A) Shown are relative expressions of ESRG, POU5F1, NANOG, T, and NES in primed H9 ESCs transduced with empty vector (shNC), and shRNAs against ESRG (2, 4, and 5). Values are normalized by GAPDH or ACTB and compared with the primed H9 ESC line. $^*$P<0.05 vs. primed H9 ESC line by unpaired t-test. n = 3. (B) Representative images of ESRG KD cells of immunocytochemistry for NANOG. Bars, 200 μm. Numerical values for A are available in S1 Data. (TIF)

**S6 Fig. Knockdown of HERV-Hs did not induce differentiation of human PSCs.** (A) The KD efficiencies of pan HERV-Hs. Shown are relative expressions of pan HERV-Hs and ESRG in primed N18 iPSCs transduced with empty vector (Mock), and shRNAs against HERV-Hs (1, 2 and 3). Values are normalized by GAPDH and compared with the primed N18 iPSC line. $^*$P<0.05 vs. primed N18 iPSC line by unpaired t-test. n = 3. (B) The expression of PSC and differentiation markers in HERV-H KD cells. The heatmap shows the normalized intensity of the indicated genes analyzed by microarray. Each value is the average of biological triplicates. (C) Representative images of HERV-H KD cells of immunocytochemistry for NANOG. Bars, 200 μm. Numerical values for A and B are available in S1 Data. (TIF)

**S1 Table. Summarized phastCons conservation scores and proportion of singletons across lincRNAs.** (XLSX)

**S2 Table. Differential expression between human and chimpanzee specific for iPSC stage (interaction term cell type:species).** (XLSX)

**S3 Table. Normalized mean expression per gene in the human and chimpanzee iPSCs.** (XLSX)

**S4 Table. The number of polymorphisms and substitutions in the human ESRG.** (XLSX)

**S5 Table. Oligo DNA sequences used in this study.** (XLSX)

**S6 Table. TaqMan Assays used in this study.** (XLSX)

**S1 Data. In separate sheets, the excel spreadsheet contains the numerical values for Figs 2B, 2C, 2F, 2H, 3A, 3C, 3F, 4C, 4F, 5A, 5B, S1A, S1B, S2B, S5A, S6A and S6B.** (XLSX)

## Acknowledgments

We would like to thank M. Iwasaki, M. Koyanagi-Aoi, A. Kunitomi, K. Okita, and D. Trono for sharing materials and data, MA. Khurram, SD. Perli, S. Wang, and K. Tomoda for discussions, and Y. Kawahara, M. Lancero, and R. Hirohata for technical assistance. We are also grateful to K. Essex, K. Higashi, K. Kamegawa, M. Otsuki, M. Saito, and S. Takeshima for administrative support, and P. Karagiannis for crucial reading of the manuscript.

## Author Contributions

**Conceptualization:** Kazutoshi Takahashi, Shinya Yamanaka.

**Formal analysis:** Kazutoshi Takahashi, Chikako Okubo, Zane Kliesmete, Mari Ohnuki, Akira Watanabe, Ines Hellmann.

**Funding acquisition:** Kazutoshi Takahashi, Shinya Yamanaka.

**Investigation:** Kazutoshi Takahashi, Michiko Nakamura, Megumi Narita.

**Methodology:** Kazutoshi Takahashi, Mai Ueda, Yasuhiro Takashima, Ines Hellmann.

**Resources:** Kazutoshi Takahashi, Mai Ueda, Yasuhiro Takashima.

**Supervision:** Kazutoshi Takahashi, Shinya Yamanaka.

**Writing – original draft:** Kazutoshi Takahashi.

**Writing – review & editing:** Kazutoshi Takahashi.

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
