## [Decision Letter · Decision Letter 0]

25 Jan 2021

Dear Dr Takahashi,

Thank you very much for submitting your Research Article entitled 'The pluripotent stem cell-specific transcript ESRG is dispensable for human pluripotency' to PLOS Genetics. We apologize for the delay in review process and have decided to proceed with a decision based on two reviews. 

The manuscript was fully evaluated at the editorial level and by independent peer reviewers. The reviewers agree that this paper is an important contribution to the field but they have raised concerns that have to be addressed, including the RNAseq replicates and the method of knockdown. Based on the reviews, we will not be able to accept this version of the manuscript, but we would be willing to review a revised version. We cannot, of course, promise publication at that time.

If you decide to revise the manuscript for further consideration at PLOS Genetics, please aim to resubmit within the next 60 days, unless it will take extra time to address the concerns of the reviewers, in which case we would appreciate an expected resubmission date by email to plosgenetics@plos.org.

[LINK]

We are sorry that we cannot be more positive about your manuscript at this stage. Please do not hesitate to contact us if you have any concerns or questions.

Yours sincerely,

Marisa S Bartolomei

Associate Editor

PLOS Genetics

Gregory Barsh

Editor-in-Chief

PLOS Genetics

Reviewer's Responses to Questions

**Comments to the Authors:**

Reviewer #1: In this manuscript, the authors demonstrated that the pluripotent stem cell-specific lncRNA ESRG is dispensable for human PSC self-renewal and lineage commitment, and reprogramming of neural stem cells to iPSCs.

It has been previously shown that ESRG knockdown causes hESC differentiation and overexpression enhanced reprogramming efficiency (Wang, et al., Nature, 2014). The authors demonstrated data that contradict with this previous report in Nature by using total deletion of ESRG gene with two gRNAs together with Cas9.

The data are solid and it is unfortunate that the authors needed to spend time to demonstrate it was ‘not’ important. It might be interesting to test exactly the same shRNA used by Wang, et al. to see its effect on PSC self-renewal and overexpression of ESRG in fibroblast reprogramming as Wang, et al. did. It is also interesting to see the pan-HERV-Hs shRNAs cause hPSC differentiation as another paper reported (Lu, Nat Struct Mol Biol, 2014), in comparison with shLTR7-1 which the authors previously used to revert the differentiation defective iPSC phenotype (Ohnuki, PNAS, 2014). These experiments could confirm whether the previously reported importance of HERV-Hs for hPSCs identity is true or not in these authors’ hands/culture condition. If the bulk HERV-Hs expression is indeed important, ESRG KO hESCs could be a useful tool to investigate how HERV-Hs control gene expression as ESRG KO hESCs have only 10 down-regulated genes? At least some of them could be directly regulated? Are any of the 10 genes closely located to ESRG and cis interaction can be observed?

Reviewer #2: In this paper Takahashi and colleagues investigate the effect of deleting the long non-coding RNA ESRG in human pluripotent stem cells. This is interesting as ESRG is driven by an HERV-H element and a good candidate to be one of the functional elements responsible for the previously observed effect on pluripotency of a general HERV-H knock-down. They find that the deletion does not result in any major pluripotency phenotype. Although this is a negative finding, I still find this result interesting and relevant as it is 1) generally important to publish negative results, 2) the analysis is overall technically sound and carefully conducted and 3) it is an “unexpected” (line 43) result from a classically biochemical/mechanistical viewpoint, but I do not find it unexpected from an evolutionary viewpoint.

However, the paper would greatly profit from some improvements:

Major concerns:

1) The most important in my view, is the analysis of the gene expression data from the 3 WT and 3 KO lines. I think 3 replicates are just not enough to infer changes in gene expression robustly. Furthermore there are apparent expression changes (Fig. 1G), but they are not discussed at all. I think a proper analysis would involve RNA-seq data and not microarray data as the false negative rate can be much better estimated for RNA-seq. Additionally, it would take ideally more clones as biological replicates or – if that is not readily available - at least independently grown replicates of the clones (e.g. 3 per clone), to have at least some power to detect expression differences in naïve and/or primed cells. The general conclusions will not be affected, as ESRG will remain dispensable for pluripotency independent of some differently expressed genes. But if ESRG is functional (See point 2 below) it will reveal insights about its potential role.

2) The second important point is to analyse the conservation of ESRG. To judge how expected or unexpected the finding of no phenotype is, it is crucial to know how much evidence is there that ESRG is indeed functional. I.e. is its promotor/gene body more conserved than expected and more or less conserved than other HERV-H elements? Maybe one could even say something about the conservation of its expression in other species from published datasets. In any case, without the most important evidence for the functionality of a genetic element this paper remains very incomplete.

More minor concerns

1) The language is not appropriate as there are way too many errors and/or imprecise usage of language that considerably weakens the impression of precise experiments. Examples include:

a. Line 102 ff: “that flanked ~8,400 bp of the genomic region including the entire ESRG gene based on the human genome database and RNA-seq data”

b. Line 107: “in 3 WT versus 3 KO manner”

c. Line 44: “contribute to reprogram of differentiated cells to pluripotent state”

2) At the beginning of the results it is not clear which data was generated for this study and how this was done and which data was already published.

**Have all data underlying the figures and results presented in the manuscript been provided?**

Reviewer #1: Yes

Reviewer #2: Yes

PLOS authors have the option to publish the peer review history of their article (what does this mean?). If published, this will include your full peer review and any attached files.

Reviewer #1: **Yes: **Keisuke Kaji

Reviewer #2: No

---

## [Decision Letter · Decision Letter 1]

6 May 2021

Dear Dr Takahashi,

We are pleased to inform you that your manuscript entitled "The pluripotent stem cell-specific transcript ESRG is dispensable for human pluripotency" has been editorially accepted for publication in PLOS Genetics. Congratulations!

Yours sincerely,

Marisa S Bartolomei

Associate Editor

PLOS Genetics

Gregory Barsh

Editor-in-Chief

PLOS Genetics

Comments from the reviewers (if applicable):

Reviewer's Responses to Questions

**Comments to the Authors:**

Reviewer #1: The authors answers to all my concerns.

**Have all data underlying the figures and results presented in the manuscript been provided?**

Reviewer #1: Yes

PLOS authors have the option to publish the peer review history of their article (what does this mean?). If published, this will include your full peer review and any attached files.

Reviewer #1: No

**Data Deposition**

http://datadryad.org/submit?journalID=pgenetics&manu=PGENETICS-D-20-01798R1

**Press Queries**

---

## [Editor Report · Acceptance letter]

21 May 2021

PGENETICS-D-20-01798R1 

The pluripotent stem cell-specific transcript ESRG is dispensable for human pluripotency 

Dear Dr Takahashi, 

We are pleased to inform you that your manuscript entitled "The pluripotent stem cell-specific transcript ESRG is dispensable for human pluripotency" has been formally accepted for publication in PLOS Genetics! Your manuscript is now with our production department and you will be notified of the publication date in due course.

With kind regards,

Katalin Szabo

PLOS Genetics

On behalf of:
